# The high optical brightness of the BlueWalker 3 satellite

Sangeetha Nandakumar[1 ✉], Siegfried Eggl[2,3 ✉], Jeremy Tregloan-Reed[1,3 ✉], Christian Adam[4], Jasmine Anderson-Baldwin[5], Michele T. Bannister[3,6], Adam Battle[7], Zouhair Benkhaldoun[3,8], Tanner Campbell[7], J. P. Colque[4], Guillermo Damke[3,9], Ilse Plauchu Frayn[10], Mourad Ghachoui[8], Pedro F. Guillen[10], Aziz Ettahar Kaeouach[11], Harrison R. Krantz[12], Marco Langbroek[13], Nicholas Rattenbury[5], Vishnu Reddy[7], Ryan Ridden-Harper[6], Brad Young[3], Eduardo Unda-Sanzana[4], Alan M. Watson[14], Constance E. Walker[3,9], John C. Barentine[3,15,16], Piero Benvenuti[3,17], Federico Di Vruno[3,18], Mike W. Peel[3,19,20,21], Meredith L. Rawls[3,22], Cees Bassa[23], Catalina Flores-Quintana[24,25], Pablo García[26,27], Sam Kim[28,29], Penélope Longa-Peña[4], Edgar Ortiz[30], Ángel Otarola[31], María Romero-Colmenares[1], Pedro Sanhueza[9], Giorgio Siringo[31,32] & Mario Soto[1]

Large constellations of bright artificial satellites in low Earth orbit pose significant challenges to ground-based astronomy[1]. Current orbiting constellation satellites have brightnesses between apparent magnitudes 4 and 6, whereas in the near-infrared Ks band, they can reach magnitude 2 (ref. 2). Satellite operators, astronomers and other users of the night sky are working on brightness mitigation strategies[3,4]. Radio emissions induce further potential risk to ground-based radio telescopes that also need to be evaluated. Here we report the outcome of an international optical observation campaign of a prototype constellation satellite, AST SpaceMobile's BlueWalker 3. BlueWalker 3 features a 64.3 $m^2$ phased-array antenna as well as a launch vehicle adaptor (LVA)[5]. The peak brightness of the satellite reached an apparent magnitude of 0.4. This made the new satellite one of the brightest objects in the night sky. Additionally, the LVA reached an apparent V-band magnitude of 5.5, four times brighter than the current International Astronomical Union recommendation of magnitude 7 (refs. 3,6); it jettisoned on 10 November 2022 (Universal Time), and its orbital ephemeris was not publicly released until 4 days later. The expected build-out of constellations with hundreds of thousands of new bright objects[1] will make active satellite tracking and avoidance strategies a necessity for ground-based telescopes.

On 10 September 2022, the AST Space Mobile prototype BlueWalker 3 (hereafter BW3) satellite was launched into orbit to test communicating directly with unmodified mobile phones using a large phased-array antenna. To ascertain the impact of BW3 to astronomy and the night sky, an international observing campaign was conducted using both amateur visual observations and professional astronomers from Chile, the USA, Mexico, Aotearoa New Zealand, The Netherlands and Morocco. Visual observations made before BW3 deployed its antenna

on 10 November 2022 (Universal Time (UT)) implied that this satellite would be particularly bright. A telescopic observation campaign confirmed visual observations[7,8], suggesting that once the antenna deployment was completed, the brightness of BW3 jumped from apparent V-band magnitudes of about +6 ± 0.3 to +0.4 ± 0.1 (Figs. 1 and 2): as bright as Procyon and Achernar, the brightest stars in the constellations of Canis Minor and Eridanus, respectively. For comparison, the unaided eye at a dark sky site will see stars of magnitude +6 (ref. 9),

[1]Instituto de Investigación en Astronomía y Ciencias Planetarias, Universidad de Atacama, Copiapó, Chile. [2]Department of Aerospace Engineering/Department of Astronomy, University of Illinois at Urbana–Champaign, Champaign, IL, USA. [3]IAU Centre for the Protection of the Dark and Quiet Sky from Satellite Constellation Interference, Paris, France. [4]Centro de Astronomía (CITEVA), Universidad de Antofagasta, Antofagasta, Chile. [5]The University of Auckland, Auckland, New Zealand. [6]School of Physical and Chemical Sciences Te Kura Matū, University of Canterbury, Christchurch, New Zealand. [7]Lunar and Planetary Laboratory, University of Arizona, Tucson, AZ, USA. [8]Oukaimeden Observatory, High Energy Physics and Astrophysics Laboratory, FSSM, Cadi Ayyad University, Marrakesh, Morocco. [9]NSFs NOIRLab, Tucson, AZ, USA. [10]Instituto de Astronomía, Universidad Nacional Autónoma de Méxic, Ensenada, Mexico. [11]High Atlas Observatory, Oukaimeden Observatory, Oukaimeden, Morocco. [12]University of Arizona Steward Observatory, Tucson, AZ, USA. [13]Faculty of Aerospace Engineering, Delft Technical University, Delft, The Netherlands. [14]Instituto de Astronomía, Universidad Nacional Autónoma de México, Mexico City, Mexico. [15]Dark Sky Consulting, LLC, Tucson, AZ, USA. [16]Consortium for Dark Sky Studies, University of Utah, Salt Lake City, UT, USA. [17]Department of Astronomy, University of Padova, Padova, Italy. [18]SKA Observatory, Jodrell Bank, Macclesfield, UK. [19]Instituto de Astrofísica de Canarias, La Laguna, Spain. [20]Departamento de Astrofísica, Universidad de La Laguna, La Laguna, Spain. [21]Imperial College London, London, UK. [22]Department of Astronomy/DiRAC/Vera C. Rubin Observatory, University of Washington, Seattle, WA, USA. [23]ASTRON Netherlands Institute for Radio Astronomy, Dwingeloo, The Netherlands. [24]Instituto de Astrofísica, Universidad Andrés Bello, Santiago, Chile. [25]Instituto Milenio de Astrofísica MAS, Santiago, Chile. [26]Instituto de Astronomía, Universidad Católica del Norte, Antofagasta, Chile. [27]Chinese Academy of Sciences South America Center for Astronomy, National Astronomical Observatories, CAS, Beijing, China. [28]Astro-Engineering Center (AIUC), Pontificia Universidad Católica de Chile, Santiago, Chile. [29]Max-Planck-Institut für Astronomie, Heidelberg, Germany. [30]Chilean Low Earth Orbit satellite (CLEOsat) Group, Universidad de Atacama, Copiapó, Chile. [31]European Southern Observatory (Chile) Alonso de Córdova, Vitacura, Chile. [32]Joint ALMA Observatory, Alonso de Córdova, Vitacura, Chile. ✉e-mail: an.sangeetha@gmail.com; eggl@illinois.edu; jeremy.tregloan-reed@uda.cl

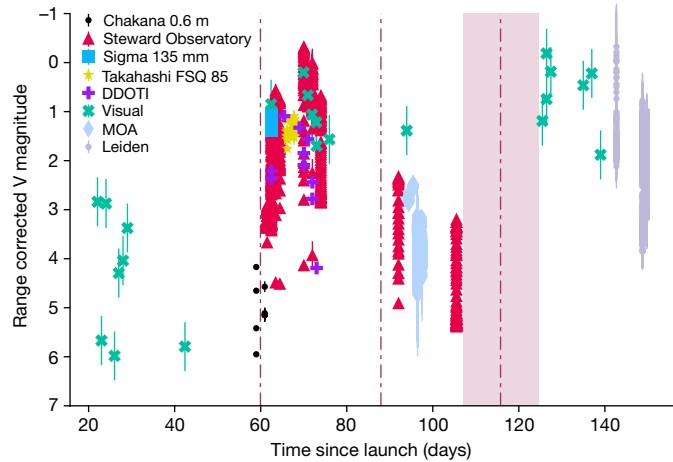

**Fig. 1 | The reflective brightness of BW3 as a function of days since launch.** The data points represent the range-corrected magnitudes, where the brightness was scaled using an inverse square law of the range versus orbital height, to provide numerical values 'as seen' at local zenith. The shape and colours represent the different observations: Ckoirama Observatory's Chakana telescope (black circles), Steward Observatory SSA astrograph (red triangles), complementary metal–oxide–semiconductor (CMOS) camera Sigma 135 mm lens (light blue squares), Oukaïmeden Observatory Takahashi FSQ 85 (yellow stars), Deca-Degree Optical Transient Imager (DDOTI; purple plus signs), visual (green crosses), MOA-II (grey-blue diamonds) and Leiden (grey circles). The vertical brown dot-dash lines from left to right are the times corresponding to the unfolding of the array (60 days since launch), the beginning of dimming (85 days since launch)[8] and our estimate of the start of brightening (117.75 ± 8.75 days since launch) based on the observations, where the shaded region denotes the uncertainty because of observing cadence. In some cases, the error bars are smaller than the data point size and represent the Poisson distribution of the integrated flux of the satellite trail in the image. All uncertainties are 1 s.d.

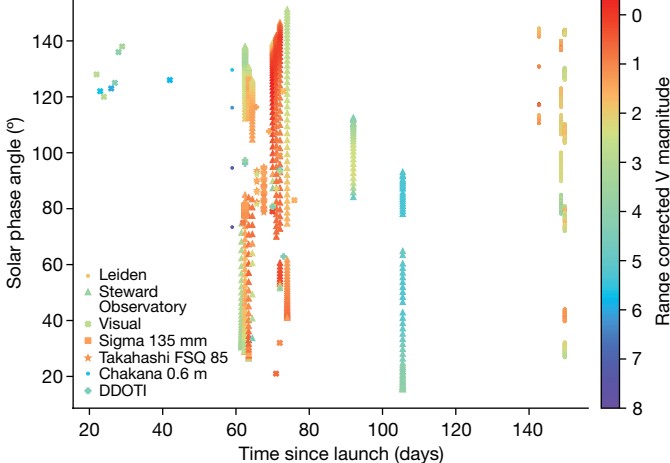

**Fig. 2 | Range-corrected V-band magnitude of BW3 as a function of the solar phase angle and days since launch.** This observation subset focuses on the unfolding event and includes a wide range of phase angles. The unfolding array is the primary contributor to the increasing reflective brightness of the satellite. The symbols represent the different observations and are the same as in Fig. 1.

reducing to approximately +2 in inner city sites. As with all satellites, the apparent brightness is not constant and changes with solar phase angle and where in the sky it is observed (for example, Fig. 3). Observations of BW3 from 8 December 2022 (UT) confirm that the satellite started to dim from $V \approx 1$ to $V \approx 6$ by 25 December 2022 (UT), likely a result of changes in attitude (orientation)[10]. However, within 3 weeks, the satellite became brighter than before, and on 3 April 2023 UT, it was as bright as magnitude +0.4. Optical observations confirm that BW3 increases in brightness when BW3 is at a higher elevation above the horizon and indicate that the range between the observer and BW3 is a primary contributor to the apparent/observed magnitude (Fig. 3). The apparent brightness of BW3 also shows correlation with solar phase angle and appears brighter at high phase angles. During the deployment process of BW3, we observed a bright object decoupling from the satellite in three datasets from the Ckoirama Chakana 0.6 m telescope combined with a simultaneous observation using the Cerro Tololo Inter-American Observatory (CTIO) 0.9 m telescope on the evening of 10 November 2022 UT. This was later identified through two-line element (TLE) matching as the launch vehicle adaptor (LVA) shown in Fig. 4. The LVA is used to house and protect the folded antenna array during launch and until time of deployment (unfolding). The apparent V-band magnitude of the LVA was measured to be around 5.5 (slightly fainter than the Starlink Gen1 satellites[3,11]): roughly four times brighter than the Dark and Quiet Skies II recommendation that the maintained brightness of a satellite should not exceed magnitude 7 when the orbital height is equal to or less than 550 km (refs. 3,6). This recommendation is specified to avoid the most severe effects on sensitive astronomical detectors[3]. After separation, LVAs and other launch-associated hardware are often left to drift for extended periods of time until they deorbit.

In the case of the LVA of BW3, it took approximately 4 days before it was listed in the public satellite catalogue[12] with orbit information. This poses additional challenges to mitigation efforts by ground-based observatories because satellite avoidance requires a complete and highly accurate set of satellite orbits.

Satellites, such as BW3, could also present an additional source of noise for radio astronomy by increasing interference in wideband receivers and potentially affecting nearby protected radio astronomy bands. Even for telescopes protected by radio quiet zones, the protections are afforded only for terrestrial transmitters. This can be a particular issue for telescopes like the Green Bank Telescope and the Square Kilometre Array (SKA), which observe at or close to frequencies also used by mobile phones, as well as those observing at higher frequencies. Along with transmissions at mobile phone frequencies (in the 750–900 MHz range), the gateway downlink frequencies for BW3 are 37.5–42.0 and 42.0–42.5 GHz (ref. 13), which are adjacent to the radio astronomy (RA) protected 42.5–43.5 GHz band. The planned launch of hundreds[14] of similar satellites over the next decade requires research into strategies to protect upcoming ground-based telescopes and surveys, including the European Southern Observatory's Extremely Large Telescope, the Las Campanas Observatory Giant Magellan Telescope and the Vera C. Rubin Observatory's Legacy Survey of Space and Time[15], and radio observatories, such as the SKA[16], the Next Generation Very Large Array[17] and the Atacama Large Millimeter Array[18].

A subset of the observations used to measure the reflective brightness of BW3 provides an opportunity to measure the TLE accuracy used in determining BW3 orbital ephemerides. When at least one start/end point of a satellite trail is within an image, it is possible to determine the satellite's exact position as a function of time from integrating the satellite sky-projected angular velocity over the trail length. When compared with the ephemerides, we find that the mean total error of BW3 from its predicted position is 7.2 arcmin (with an s.d. of ±3.1 arcmin) and is the quadrature sum of both the spatial and temporal errors (the temporal error is translated to a spatial error after scaling by the sky-projected angular velocity), with mean values of 72.4 arcsec and 1.7 s, respectively. However, because of atmospheric drag, solar activity (influences atmospheric scale height) and orbital dynamics, the accuracy of a TLE degrades over time. When the TLE accuracy measurements are compared with the epochs of the different TLEs, the decay rate is measured to be 0.4 ± 0.2 arcmin h⁻¹ (although we note that the decay rate will strongly depend on solar activity and

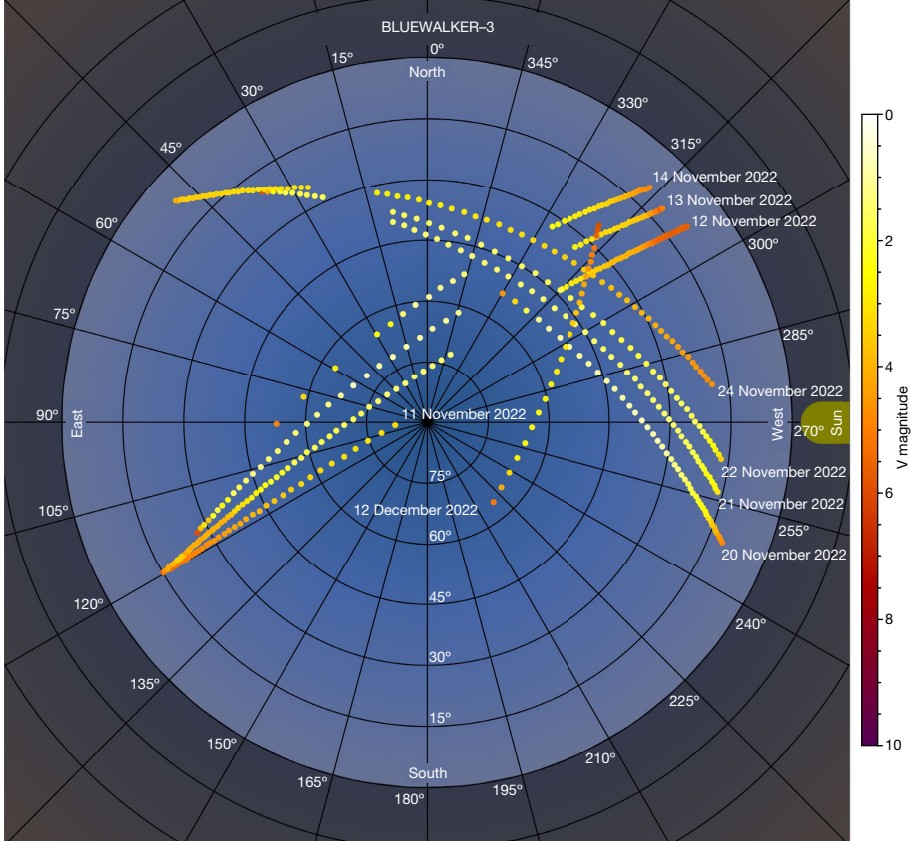

**Fig. 3 | BW3 apparent brightness all-sky plot from nine passes observed from Steward Observatory.** The apparent brightness of BW3 after unfolding as observed by the Steward Observatory SSA astrograph on nine different passes on nine different nights in November and December 2022 (11–14 November, 20–22 November, 24 November and 12 December UT). This all-sky plot shows the correlation of apparent brightness with the on-sky position of the satellite relative to the Sun (yellow mark on the right). To standardize the Sun's azimuth, which is not the same for all observations, the plotted position of each point is rotated around zenith such that the below-horizon Sun is at the same azimuth (west) for all measurements.

orbital altitude). This highlights the importance of regularly updated TLEs in short time intervals (for example, Starlink supplemental TLEs are updated every 8 h).

Despite many efforts by the aerospace industry, policy makers, astronomers and the community at large to mitigate the impact of these satellites on ground-based astronomy[1–4,11], with individual examples

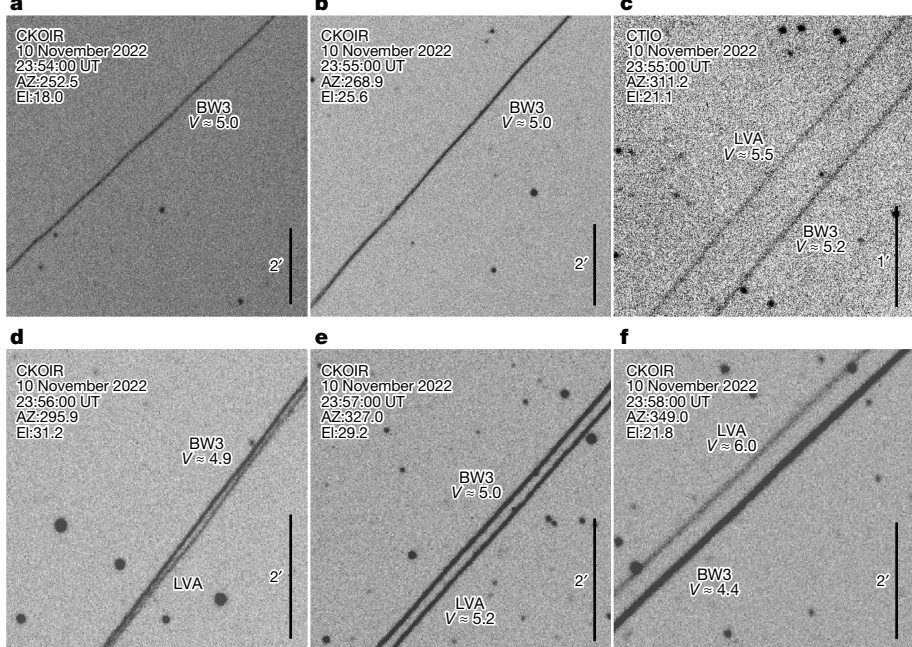

**Fig. 4 | Optical observations of BW3 and its associated LVA taken on 10 November 2022 (UT).** The images are subframes of the full Flexible Image Transport System (FITS) images. **a–f**, Images were taken simultaneously using the Chakana (CKOIR) 0.6 m (**b**) and CTIO 0.9 m (**c**) telescopes at 23:55:00 UT, whereas the first observation (**a**) and final three observations (**d–f**) were taken with the Chakana 0.6 m telescope at 23:54:00 UT (**a**) and at 23:56:00 UT (**d**), 23:57:00 UT (**e**) and 23:58:00 UT (**f**), respectively, with the final three observations showing increasing angular separation. The measured range-corrected magnitudes are provided for both BW3 and the LVA, with the exception of **d**. Because of the two tracks being very close to each other and in some cases, overlapping, it was not possible to measure the individual magnitudes of each track. The magnitude of $V \approx 4.9$ given for BW3 (**d**) is effectively the total brightness of both tracks. The azimuth (AZ) and elevation (El) of each observation is given in the upper left corners of each image.

such as the Starlink Darksat and VisorSat mitigation designs and Bragg coatings on Starlink Gen2 satellites, the trend toward the launch of increasingly larger and brighter satellites continues to grow. Impact assessments for satellite operators before launch could help ensure that the impact of their satellites on the space and Earth environments is critically evaluated. We encourage the implementation of such studies as part of launching authorization processes.

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

# Article

## Methods

### Observations and data reduction

Observations are reported per observatory below. The observatories and the corresponding orbit of BW3 on the days of the observations are shown in Extended Data Fig. 1. All reported satellite magnitudes are corrected for exposure time and the satellite's angular speed across the sky. For images where the trail has no start or end points, this is determined by the ratio of the time required to transverse the detector, $\tau$, based on the trail length and the satellite's angular velocity, dividing by the exposure time, $t$. The integrated flux of the background stars is then multiplied by this ratio. This is required because although the integrated flux of the background stars is total flux received in $t$ seconds, the total flux within the trail aperture is received in $\tau$ seconds.

### CMOS Sigma 0.135 m observations

BW3 was observed on two nights (12 and 13 November 2022 UT) from Tucson, Arizona, USA (latitude 32.3740°, longitude −111.0168°) using a ZWO 432 complementary metal–oxide–semiconductor (CMOS) sensor attached to a Sigma 135 mm f/1.8 lens. The weather was photometric on the 12th and high cirrus on the 13th; therefore, only the data from the 12th are reported. The data were collected by pointing the instrument at a star along the predicted track of the satellite. To avoid saturation, exposure times for BW3 were kept short (16 ms), which created an additional challenge of insufficient signal-to-noise ratio (SNR) on background stars for photometric calibration. Five 15 s sidereal-tracked images of the same field were obtained before and after the satellite pass (10 images total) for photometric calibration.

The linear photometric correction used to convert from instrumental magnitude to GAIA G magnitude was calculated by using all solar-type stars identified in the set of images after iterative sigma clipping, weighted by the individual flux measurement uncertainties (that is, reciprocal of SNR). Solar-type stars are identified using the GAIA DR2 (ref. 19), and when multiple measurements of the same star are found across multiple images, a median flux and SNR are used in the fit. The root-mean-square (r.m.s.) uncertainty quoted was calculated as the r.m.s. error between the postcorrection measured magnitudes of all solar-type stars in the images (no sigma clipping) and their published GAIA DR2 magnitudes. The Johnson V-band magnitudes were then converted from the GAIA G magnitudes[20].

### Chakana, CTIO, DDOTI and Oukaïmeden observations

Observations of BW3 were performed on two nights, 9 and 10 November 2022 UT, using the Chakana 0.6 m telescope located at the Ckoirama Observatory[21] in northern Chile (latitude −24.089°, longitude −69.931°). Observations used the Sloan g′ filter and a 2 × 2 binning technique to reduce the charge-coupled device (CCD) readout dead time and increase the SNR of background stars, thereby improving the accuracy of the differential magnitude measurements of the satellite trail. The observing strategy used the standard 'wait and catch' technique while tracking at the sidereal rate, where the telescope was positioned at the predicted RA and declination (dec.) derived from a low Earth orbit (LEO) satellite ephemeris code (https://github.com/CLEOsat-group/satellite-tracking), and the exposure began 5 s before BW3 reached the predicted coordinates.

Observations of BW3 were also obtained on the night of 10 November 2022 UT using the CTIO's SMARTS 0.9 m telescope located in northern Chile (latitude −30.169°, longitude −70.806°). The imager consists of a Tek2K 2,048 × 2,048 pixels CCD detector, with a 0.401 arcsec pixel⁻¹ scale and a 13.6 arcmin field of view. The observations were obtained in the Johnson V band. Similarly to observations from Ckoirama Observatory, a wait and catch technique was adopted tracking at sidereal rate. We opened the imager shutter 10 s before the pass and obtained a 20 s exposure to capture the satellite trail. This pass image was followed by a 20 s comparison image of the same field in the same filter band.

Observations of BW3 were also conducted on 12, 14, 15, 18, 19, 20, 21, 22 and 23 November 2022 UT with the DDOTI wide-field imager[22] at the Observatorio Astronómico Nacional on the Sierra de San Pedro Mártir, Baja California, Mexico (latitude 31.0455°, longitude −115.4658°). DDOTI has six 28 cm telescopes on a common mount. Each telescope is equipped with an unfiltered 6,000 × 6,000 CCD, which gives a field of 3.4° × 3.4° and a pixel scale of about 2 arcsec pixel⁻¹. For the observations presented here, two of the telescopes were out of service for maintenance, and so, the total field was about 7° × 7°.

Further observations of BW3 were conducted on 16 and 17 November 2022 UT with a Takahashi telescope in Oukaïmeden, Morocco (latitude 31.2064°, longitude −7.866°)[23]. The observations used a CMOS camera and performed both 2 s exposures (16 November 2022) and 1 s exposures (17 November 2022). Sidereal tracking was used, and the data allow for the extraction of the reflective brightness of BW3 and estimate the accuracy of the TLE used to predict the sky position.

The reduction, astrometric calibration, satellite trail detection and its analysis for the observations from Chakana, DDOTI and Oukaïmeden were performed with the CLEOSat pipeline, a custom and open-source end-to-end Python pipeline for the processing and analysis of satellite trail observations (C.A., J.T.-R., E.U.-S. and S.N., manuscript in preparation).

The raw FITS image files were processed with ccdproc (https://ccdproc.readthedocs.io/en/). This includes subtracting the dark frame from each image to remove the instrumental signature as well as dividing by the average flat field to correct for non-uniform sensitivity across the chip.

The reduced FITS images were then calibrated to be able to map the positions of sources on the detector to their celestial coordinates on the sky. The positions of sources on the detector were extracted using photutils (https://photutils.readthedocs.io/en/stable/). For reference, a catalogue of stellar sources, with known and precise positions on the sky, was compiled from the GAIA DR3 (refs. 24,25) catalogue via astroquery[26]. The required transformation (that is, the scale, rotation and translation to match both coordinate systems) was then determined by applying a phase correlation algorithm to the distances and angles calculated between each source from both catalogues to determine the location of the peak in the cross-correlation spectrum yielding the correction to the World Coordinate System information in the FITS header.

To identify the satellite trail(s) in the observations, first a sharpen filter was applied to the image to increase contrast and facilitate detection of fainter trails. Then, a source detection algorithm was used to create a segmentation map containing all sources, including trails, $1\sigma$ above the background in the sharpened image. The resulting segments were filtered, retaining only the most eccentric segments ($e > 0.99$).

To identify and characterize the satellite trail(s), we followed the approach for line segment detection using the Hough transform[27]. In this approach, a voting procedure is performed that associates a set of lines in the $x-y$ image space to a pair of values in the $\Theta-\rho$ plane, also referred to as Hough space, and given a voting angle in the image space, this voting distribution is analysed. Regarding the corresponding column in Hough space, voting along the distance axis is considered as being a random variable, and voting values in cells of the discrete Hough space are considered as forming a probability distribution. The statistical characteristics of this probability distribution are used to fit a quadratic polynomial curve and a linear curve whose coefficients yielded the direction, length and width of a line segment as well as the midpoint of a line segment, respectively.

The magnitude of the trail was estimated using aperture photometry by comparing the instrumental magnitude of the trail with the well-known magnitudes of a set of comparison stars in the image[28]. To estimate the optimum aperture at which most of the light from the source is captured while minimizing contamination from the sky background and unrelated sources, first the SNR for different aperture

sizes was measured. The aperture radius at which the SNR is maximized was then multiplied by a factor to allow for any error in centroiding and was used as the new aperture radius for the image. Additionally, the magnitude correction required to compensate for the flux lost because of a finite aperture size was determined by calculating the ratio of fluxes in the optimum aperture and a larger standard aperture. The average value and s.d. of the resulting distribution of magnitude corrections were then taken as the aperture correction and applied to all sources measured with standard aperture size during aperture photometry.

### Steward Observatory observations

Figure 3 shows the apparent brightness of BW3 after deployment as observed by the Steward Observatory SSA astrograph on nine different passes on nine different nights in November and December 2022. The Steward Observatory SSA astrograph is a unique system specifically created to observe Earth-orbiting satellites and space debris[29]. The system resides in a portable trailer-mounted enclosure that was stationed at the Mt. Lemmon SkyCenter near Tucson, Arizona (latitude 32.4420°, longitude −110.7893°) for the reported observations.

The astrograph tracked BW3 as it passed overhead and continually recorded images at a rate of approximately one image every 5 s. All observations used a Johnson–Cousins $V$ filter. On the first two observed passes (11 and 12 November 2022 UT), BW3 saturated the detector, and the brightest measurements from these two passes should be considered an upper bound on the V-band magnitude (that is, the satellite was at least this bright). Subsequent observations used a shorter exposure time and were defocused to avoid saturation.

We processed the images and produced calibrated photometric measurements with a suite of software created for The Steward Observatory LEO Satellite Photometric Survey[30]. When tracking the satellite, the background stars are severely streaked, and extracting astrometric or photometric references is all but impossible. Instead, we used other sidereal observations from the same night to create an air mass extinction model and determine the photometric zero point for each observation of the tracked satellite. Figure 2 shows the correlation of apparent brightness with on-sky position and that BW3 typically appears brighter than fourth magnitude across most of the sky.

### Ōtehīwai Mt. John observations

Eight passes of BW3 were observed on 14–17 December 2022 with the 1.8 m MOA-II telescope at the University of Canterbury's Mt. John Observatory on Ōtehīwai Mt. John, Takapō, Aotearoa New Zealand (latitude −43.9857°, longitude 170.4651°). Images were acquired with sidereal tracking with MOA-cam3 (ref. 31) (1.32° × 1.65° field of view (FOV), 0.57″/ px⁻¹) in the broadband MOA-R filter (632–860 nm)[31]. Each pointing was selected based on that day's TLE from Celestrak via Heavens-Above (https://www.heavens-above.com/), with the shutter opening timed so that the satellite was predicted to cross the camera's 2.2 deg² field of view at the midpoint of the exposure. Conditions were often cloudy and windy for the entire run, with some stretches of photometric conditions with seeing of 2–3″.

Reduction of instrumental signatures for dark current and flatfielding as well as photometric calibration of field stars to SkyMapper Southern Sky Survey DR1.1 (ref. 32) (Vizier: CDS/II/358/smss) to establish zero points was made with Pouākai (https://github.com/CheerfulUser/ Pouakai), with astrometric calibration via astrometry.net (ref. 33).

Images acquired back to back were subtracted. Circular aperture photometry was applied at 0.005 s intervals along the TLE-predicted satellite trail. As the TLE did not correspond to a detected trail, we fit a Gaussian to the normalized counts of the detected trail to spatially offset the TLE onto the trail centre. MOA-R is a substantially wider band-pass filter than Johnson V band, but assuming BW3 is a solar-neutral reflector, we applied an image-specific offset to the MOA-R zero points, thereby transforming the MOA-R photometry to the Johnson V magnitudes reported here.

### Video observations from Leiden

Video observations on BW3 were conducted from Leiden, The Netherlands (latitude 52.1540°, longitude 4.4908°) on the evenings of 21, 24 and 29 September 2022; 8 December 2022; 31 January 2023; and 6 and 7 February 2023. These observations yielded both photometry and astrometry (2022 photometry previously reported[8], astrometry and 2023 photometric data are in this paper).

The camera used is a sensitive WATEC 902H2 Supreme Low Light Level CCTV camera equipped with a Pentax 1.2/50 mm lens and filming at 25 frames s⁻¹. This camera/lens combination has a 5.5° × 7.4° FOV at a scale of 35.4″ pixel⁻¹. The camera and lens combination has an average astrometric accuracy of 22.3″ ± 6.9″ (where the error is 1 s.d.) based on calibration observations on SWARM (ESO mission: The Earth's Magnetic Field and Environment Explorers) satellites (for which global navigation satellite system positions are available). Magnitudes were measured in the red band by TANGRA (http://www.hristopavlov.net/Tangra3/). These 'red' magnitudes were transformed into V-band magnitude equivalent values by mapping over 200 brightness measurements on stars against their catalogue V-band magnitude, from which an empirical relationship for 'red' magnitudes against V-band magnitudes was obtained for this camera. The Phase Alternating Line (PAL) signal output from the camera was fed into a GPSBoxsprite-2 global positioning system time inserter, which imprinted each video frame with a pulse per second time signal, allowing timekeeping at the millisecond level. The signal was next digitized by an EZcap dongle and recorded in AAV format on a laptop using OccuRec (http://www.hristopavlov.net/ OccuRec/OccuRec.html). The video frames were astrometrically solved on a frame-by-frame basis with TANGRA software (https://www.hristopavlov.net/Tangra3/) using the UCAC-4 star catalogue as a reference.

From $n$ = 4,926 astrometric observations obtained on seven separate nights, an average angular difference between the TLE-predicted positions and observed astrometric positions of 3.3 ± 0.1 (1 s.d.) arcmin was determined compared with a measured astrometric uncertainty for the instrument of 22″ ± 7″.

### Visual observations

Visual observations were made between 3 October 2022 and 16 January 2023 UT from sites near Tulsa, Oklahoma, USA. The observer has 20+ years of experience of tracking satellites, has seen 8,000+ unique objects and has reported 25,000+ passes and thousands of brightness estimates of satellites and variable stars (https://hafsnt.com/). In some cases, handheld binoculars were used. Data were gathered on 23 occasions[7]. Pass predictions were obtained for the observer's two sites (latitude +36.139°, longitude −95.983° and latitude +35.831°, longitude −96.141°) from https://www.heavens-above.com/. At a minimum, the location, expected time of observation and reliable limiting brightness must be known, bearing in mind that moving objects may appear visually dimmer than predicted. Observable passes were selected, with low elevation passes, ones in deep twilight or those at unfavourable phase angles discarded.

Suitable comparison stars were chosen to provide brightness measurements. A newly launched satellite does not always match the predictions, especially in brightness and often in timing ('TLE accuracy'). Comparison stars to estimate magnitudes were suitably bright for the stage of twilight during observations, and alternate stars were chosen in case the pass was offtrack, early, or late. All comparison star magnitudes used were obtained from the extended Hipparcos catalogue[34].

The object was observed as it passed the stars selected, so that direct comparison could be made. Any significant brightness variations, such as flashing or flaring, were also recorded at that time. Before deployment of the phased-array antenna on 10 November 2022 UT, the visual magnitude of BW3 was 6.1 ± 0.2 versus 2.4 ± 0.2 after deployment of the array. The range-corrected magnitudes were calculated using −5 log(range 500 km⁻¹) and are shown in Fig. 1, and it is the brightness

of the object when viewed with a range equal to the orbital height (that is, at local zenith with air mass = 1) at 500 km and fully lit. The uncertainties are based on experience from data gathered over decades by the observer; when comparison stars are available, accuracy in estimating visual magnitude is 0.2 mag, whereas without comparison stars, this value increases to 0.5 mag.

## LVA brightness measurement

On the evening of 10 November 2022 UT, coordinated simultaneous observations of BW3 were attempted using both the Chakana 0.6 m telescope, Ckoirama, Antofagasta, Chile (latitude −24.089°, longitude −69.931°) and the 0.9 m telescope at CTIO, Chile (latitude −30.165°, longitude 289.185° E). Because of the low elevation (less than 25°) of the apex of the forecasted sky track observed from CTIO, only a single position for which both telescopes were able to point to was selected to perform a simultaneous observation, with the exposures starting at 23:54:56.70 (UT) and 23:54:55.22 (UT) for the two telescopes, respectively. The images are given in Fig. 4 and show that from Ckoirama's viewpoint, only a single track is detected, whereas from CTIO, two tracks are seen. This provides a rare opportunity to triangulate the position of the BW3's LVA in relation to BW3.

The sky-projected angular separation in the CTIO image is measured to be 26.8 ± 5.2 arcsec. When combined with a range of 1,201 km, it provides a sky-projected separation of 78.1 ± 8.8 m. Making the assumption that the sky-projected separation lies along the line of sight from the Chakana telescope (hence, a single-track detection), we calculate the geocentric coordinates $(\hat{X}, \hat{Y}, \hat{Z})$ of both BW3 and the LVA, where the difference is found to be −55.95 m $\hat{X}$, −52.30 m $\hat{Y}$ and −15.3 m $\hat{Z}$. This value is then converted to longitude, latitude and altitude, where we find that at 23:55:00 (UT), the LVA position is latitude −24.041053°, longitude −78.672693° and Alt: 524.31 km, whereas for BW3, it is latitude −24.041074°, longitude −78.672101° and altitude (Alt): 524.27 km. As a check, we calculate the RA and dec. from both Ckoirama and CTIO for the LVA and compare these with the RA and dec. of BW3. We find that the difference in RA and dec. for Ckoirama is zero, as expected, because the LVA and BW3 follow the same track, whereas for CTIO, the difference equates to 26.8 arcsec (ΔRA = 24.8 arcsec, Δdec. = 10.2 arcsec).

Additional observations from Ckoirama, one prior and three afterward, showed that the last three images in the sequence contained two trails (Fig. 4d–f). The sky-projected angular separation is seen to be increasing; an angular separation is of 2.2 ± 1.5 arcsec at 23:56:00 (UT). Then, by 23:57:00 (UT), an angular separation of 8.2 ± 2.9 arcsec is observed, whereas by 23:58:00 (UT), this has increased to 14.8 ± 3.8 arcsec. This translates to a sky-projected separation of 9 ± 3, 19 ± 4 and 42 ± 6 m using the range between BW3 and Ckoirama of 920.31, 962.5 and 1,167.0 km, respectively, at the time of the observations. However, the first image (23:54:00 UT) shows a single track, indicating that the measured angular separation observed in the final three images is a combination of changing viewpoints, separation velocity and angular rotation velocity. The first two images from the Chakana telescope are in the west (azimuth (AZ): 252.2° and 268.9°), whereas the CTIO observation and subsequent Chakana observations, which clearly show the two trails, are toward the north and northwest (AZ: 311.2°, 327.0° and 349.0°). Without a second simultaneous observation, it is not possible to calculate the orbital velocity of the LVA and therefore, determine how much of the observed sky-projected angular separation is owing to a changing viewpoint.

## TLE accuracy

The coordinates $(\alpha, \delta)$ of the midpoint of the observed satellite trails of BW3 were obtained using the LEOSat pipeline. The pipeline uses observations and TLEs to detect and analyse satellite trails. Given the TLE, the pipeline provides magnitudes and coordinates of the trails by performing astrometric and photometric calibrations and so, measures the observed trail length. In addition, the pipeline uses the

pyorbital package (https://github.com/pytroll/pyorbital) to perform orbital calculations (using the SGP4 simplified perturbation model) to determine the RA and dec. of a satellite for given times. Measurements of the TLE spatial and temporal accuracy require datasets that contain at least one end point of the satellite trail. Satellite trails are point sources spread over a specific length because of the satellite's angular velocity. For datasets where there are no end points (that is, small FOV detectors), it is only possible to measure the spatial accuracy of the TLE because of the fact that it is not possible to precisely know the satellite's position as a function of time (that is, the point source could be at any point along the trail for any given time in the exposure timestamp).

To select coordinates along the sky track of BW3 on each pass, the BW3 TLEs used to predict BW3 orbital ephemerides and therefore, the sky position were used to provide a predicted track in the image. This provides an opportunity to measure the accuracy of the TLEs of BW3 for the observations in which at least one end point of the trail is visible. This provides a boundary condition when integrating the angular velocity over the trail length to obtain the position as a function of time of a moving source. The total error ($\sigma_{tle}$) is the quadrature sum of the spatial ($\sigma_r$) and temporal errors ($\sigma_t$):

$$\sigma_{tle} = \sqrt{\sigma_r{}^2 + \sigma_t{}^2}. \tag{1}$$

$\sigma_{tle}$ is calculated using the difference between the RA and dec. predicted by the TLE at the midpoint of the observation ($\alpha_{tle}, \delta_{tle}$) and the RA and dec. ($\alpha_{obs}, \delta_{obs}$) of BW3 in the image, when two end points are observed. When only one end of the trail is visible, the RA and dec. of this point are used instead. The data at hand show that the full mean error is $|\sigma_{tle}| = 7.2$ arcmin ± 3.1 arcmin (1 s.d.). $\sigma_r$ is simply the length of the line of intersection between ($\alpha_{obs}, \delta_{obs}$) and ($\alpha_{tle}, \delta_{tle}$), which lies perpendicular to both the observed and extrapolated TLE-predicted trails. $\sigma_t$ is then found using equation (1). This process allows the sign of $\sigma_t$ to dictate whether the predicted TLE position leads (positive) or lags behind (negative) the satellite. The results show that while below an elevation of 20°, $|\sigma_t| = 2.38 \pm 0.42$ s, where the quoted uncertainty is the 1$\sigma$ distribution. However, when above 20° elevation, this reduces to $|\sigma_t| = 0.18 \pm 0.06$ s.

To determine the confidence of the large error measured from the Takahashi data, all known instrumental uncertainties were examined. To examine the propagation of the uncertainties in the orbital equations used by pyorbital, some of the data were compared with TLE predictions using skyfield (https://pypi.org/project/skyfield/). The predicted positions between the two models differ on average by 30 arcsec, which alone cannot explain the large 24 ± 6 arcmin difference between the measured BW3 position and that predicted by the TLE. The next set of errors can come from the accuracy of the telescope location. A set of models was created with random changes in the location of the telescope by up to 150 m, and the resultant changes in the predicted sky position from the TLE were recorded and found to be on average ±1.2 arcmin. The third possible source of uncertainty is from the instrument timing uncertainty. For professional telescopes, such the DDOTI, this is less than 100 ms. If a conservative error budget of 1 s is assumed for the Takahashi data, it creates a positional uncertainty of ±10.7 arcmin. Taking these additional errors and applying them to the Takahashi data measurements give a TLE measurement accuracy of 24 ± 11 arcmin, meaning that using the worst-case instrumental, telescope position and orbital equation uncertainties, a minimum TLE accuracy is found to be greater than 13 arcmin.

The TLE errors are shown in Extended Data Fig. 2 and hint that the timing errors are dependent on elevation. For sensitive detectors, such as those of the Simonyi Survey Telescope's camera of the Vera C. Rubin Observatory, having low-accuracy ephemeris predictions of bright ($V < 7$) artificial satellites will be a major concern. It will prevent the avoidance of satellites that are too bright for minimization of the

impacts of electronic ghosts and non-linear image artefacts by correction to at least the background level[3].

Another source of error of the TLE accuracy is the amount of time passed since the TLE epoch. This is often every 8 h but can be longer in some cases. For example, once the LVA was added to the catalogue, there were about two updates a day on average, varying between zero and four per day, with occasions where there were no updates for several days. BW3 orbits are usually updated about two times per day, but sometimes, there are gaps up to 1.5 days. As the TLE provides position and velocity vectors, for a single point in time (the TLE epoch) the accuracy will depend on how far forward in time the prediction is. Therefore, by comparing the accuracy as a function of time elapsed since the different TLE epochs and the time of the observations, we determine that the decay rate of the TLE accuracy is $0.4 \pm 0.2$ arcmin h$^{-1}$. Before determining the decay rate, we performed sigma clipping to remove outliers (the Takahashi data). Because the majority of astronomical observations take place above 30° elevation combined with the lower elevation (less than 20°) and therefore, larger error measured by the Takahashi data, we ignore these data in measuring the decay rate to prevent overestimating the decay rate (Extended Data Fig. 2).

## Data availability

Raw fits images, including calibration frames, are available on Zenodo (https://doi.org/10.5281/zenodo.8102655). Raw video data (AAV format) are available on Zenodo (https://doi.org/10.5281/zenodo.8102655). The visual data of B.Y. are available at https://hafsnt.com/index.php/blue-walker-3/. The astrometric data of M.L. are available at https://doi.org/10.4121/07711969-6e9a-4944-9132-8cdc005ee6a4. Reduced data tables are provided as supplementary machine readable files. Source data are provided with this paper.

## Code availability

Prediction of expected RA and dec. of satellite for observation is from https://github.com/CLEOsat-group/satellite-tracking. Processing raw FITS files is from https://ccdproc.readthedocs.io/en/. Detection of sources for astrometric calibration is at https://photutils.readthedocs.io/en/stable/. Reduction of Ōtehīwai Mt. John observations is at https://github.com/CheerfulUser/Pouakai. TLE accuracy is at https://github.com/pytroll/pyorbital and https://pypi.org/project/skyfield/. LEOSat pipeline magnitudes and positions of satellite tracks are at https://github.com/CLEOsat-group/leosatpy https://zenodo.org/record/8012132 and https://doi.org/10.5281/zenodo.8012132. TLEerror code is at Zenodo: https://zenodo.org/records/8132639.

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

**Acknowledgements** S.E. and C.E.W. acknowledge the support of the IAU CPS SatHub. S.E. acknowledges D. Stanley for his assistance with satellite ground track generation. J.T.-R. acknowledges financial support from ANID/FONDECYT (an Initiation in Research grant; Project Number 11220287), and this material is based upon work supported by the Air Force Office of Scientific Research (Award FA9550-22-1-0292). C.A. acknowledges the support of the Regional Fund granted by the ESO–Government of Chile Joint Committee. J.A.-B. and M.T.B. acknowledge F. Gunn and I. Bell-Butler for their support in the Ōtehīwai Mt. John observations. M.T.B. appreciates support from the Rutherford Discovery Fellowships from New Zealand Government funding, administered by the Royal Society Te Apārangi. J.P.C. and E.U-S. acknowledge the assistance of M. Rocchetto and S. Fossey to set up Ckoirama. H.R.K. acknowledges the Steward Observatory Mountain Operations staff for their support at the Mt. Lemmon SkyCenter facility. E.O acknowledges support from the National Agency for Research and Development (ANID)/Scholarship Program/DOCTORADO BECAS NACIONAL CHILE/2018-21190387. Some of the data used in this paper were acquired with the DDOTI instrument at the Observatorio Astronómico Nacional on the Sierra de San Pedro Mártir, Baja California, Mexico. DDOTI is funded by CONACyT (Grants LN 232649, LN 260369, LN 271117 and 277901), the Universidad Nacional Autónoma de México (Grants CIC and DGAPA/PAPIIT IG100414, IT102715, AG100317, IN109418, IG100820 and IN105921), the NASA Goddard Space Flight Center and the University of Maryland (Grant NNX17AK54G). DDOTI is operated and maintained by the Observatorio Astronómico Nacional and the Instituto de Astronomía of the Universidad Nacional Autónoma de México. We acknowledge the contribution of N. Gehrels to the development of DDOTI. Part of this work is based on observations at Cerro Tololo Inter-American Observatory, National Science Foundation's NOIRLab, which is managed by the Association of Universities for Research in Astronomy under a cooperative agreement with the National Science Foundation. This research used data from the SMARTS 0.9m telescope, which is operated as part of the SMARTS Consortium. We acknowledge the substantial time and effort required to carry out this observing campaign, which was voluntarily contributed.

**Author contributions** M.L.R., M.W.P., C.E.W. and F.D.V. are responsible for the concept of this work. S.N. wrote the main manuscript and reduced and analysed the TLE data. S.E., M.W.P., C.E.W., F.D.V., P.B. and J.T.-R. supervised the project. J.T.-R. organized and planned the observations from Chakana, DDOTI and Oukaimeden observatories, and J.T.-R. performed LVA analysis and wrote the LVA section. B.Y. authored the section on visual observations and contributed to the data submission process. H.R.K. authored the section describing the Steward Observatory Observations, created Fig. 4 and provided the corresponding measurements. V.R. authored the section describing the CMOS Observations and created Fig. 3. A.B., T.C. and V.R. conducted CMOS observations and provided the corresponding measurements. M.T.B., J.A.-B. and N.R. planned and conducted the Mt. John Observatory observations and authored the corresponding section. R.R.-H. conducted the data analysis. E.O. developed the satellite tracking code used to plan observations at Ckoirama, DDOTI, CTIO 0.9m and Oukaimeden observatories. The satellite tracking code evolved from an original script provided by A.O. supervised by J.T.-R. at CLEOSat. C.A. developed the satellite analysis pipeline employed for the reduction, astrometric calibration, detection and photometry of the observations from the Ckoirama, DDOTI, CTIO 0.9m and Oukaimeden observatories. Z.B., M.G. and A.E.K. carried out the observations in the Oukaimeden Observatory. C.A., J.P.C. and E.U.-S. carried out the observations in the Ckoirama Observatory. M.L. carried out observations from Leiden, conducted the astrometric and photometric processing of those observations and authored the corresponding part of Methods. A.M.W., I.P.F. and P.F.G. carried out the observations with DDOTI. A.M.W. wrote the corresponding part of Methods. G.D. performed the CTIO observation. S.N., J.T.-R., M.W.P., S.E. and M.L.R. led the responses to the referee comments. All authors discussed the results and implications, and they all commented on the manuscript at all stages.

**Competing interests** The authors declare no competing interests.

**Additional information**
**Correspondence and requests for materials** should be addressed to Sangeetha Nandakumar, Siegfried Eggl or Jeremy Tregloan-Reed.

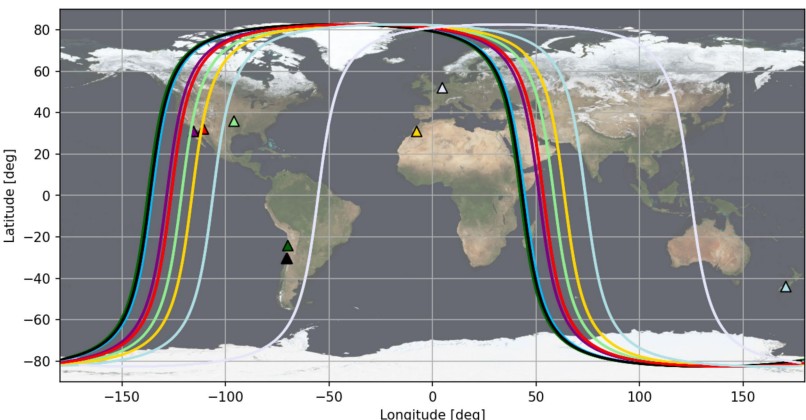

**Extended Data Fig. 1 | Observer locations and BW3 orbits.** Locations of BW3 orbits (lines) and observer locations (triangles), with timing and colour-coding to match the observations reported in Fig. 1.

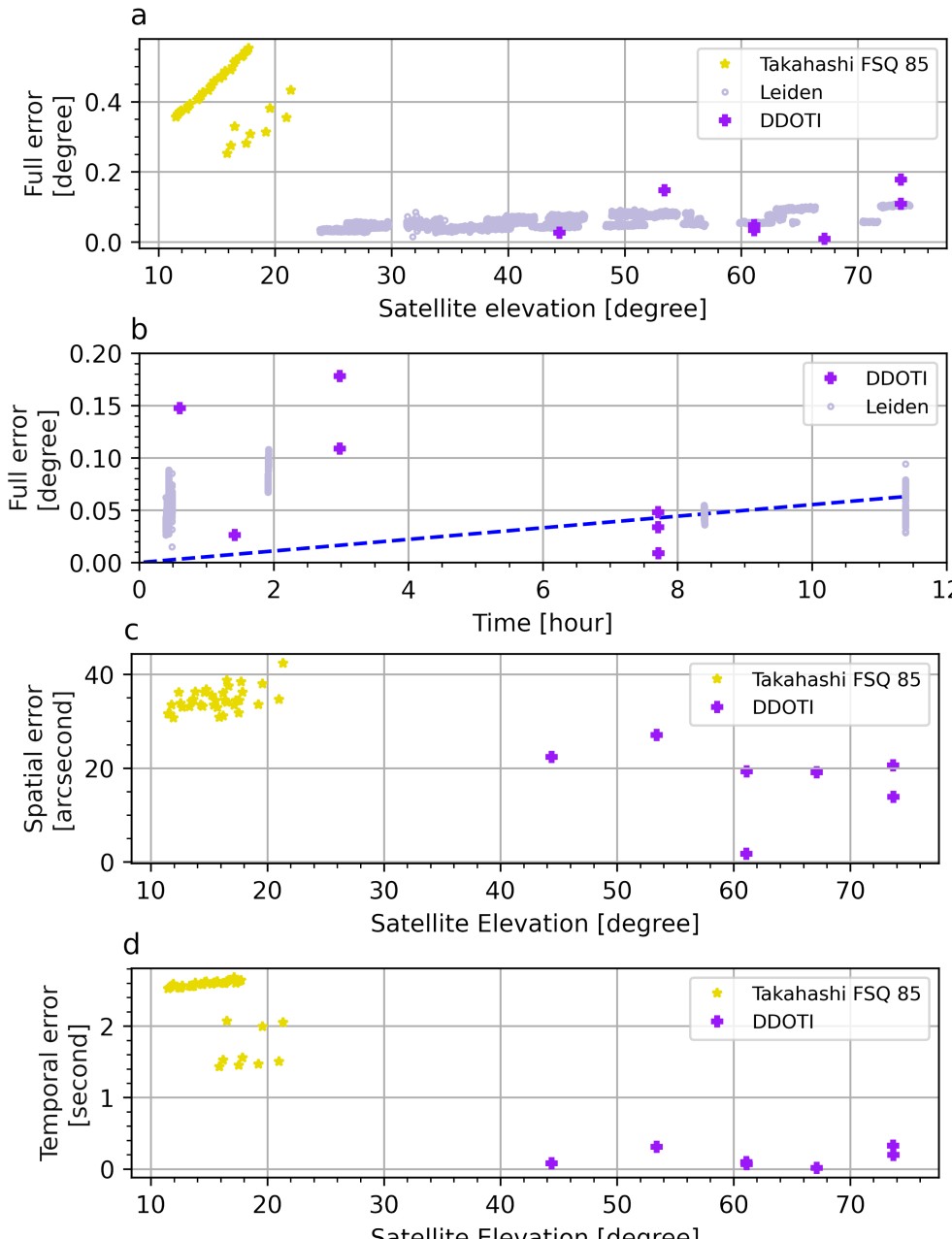

**Extended Data Fig. 2 | (a–d) Results from the TLE analysis of BW3.** The full error as measured from the difference between the measured position of the satellite trail and that forecasted by the TLE (a), the full error as a function of time between the TLE epoch and the observation (b), the spatial error, which is the measure of the perpendicular distance between the observed and TLE forecasted trails (c), and the temporal error (d). A linear trend is observed showing an accuracy decay rate of 0.4 ± 0.2 arcmin hr-1, when the Takahashi data is excluded (blue dashed line, Extended Data Fig. 2b). The DDOTI data show a low temporal error, but due to higher elevations and so larger angular velocity, these small timing errors translate to larger full errors.