## [Peer Review File · Nature]

Manuscript Title: The high optical brightness of the BlueWalker 3 satellite

Reviewer Comments & Author Rebuttals

Reviewer Reports on the Initial Version:

Referees' comments:

Referee #1 (Remarks to the Author):

Summary: The challenge of satellite constellations is an acknowledged issue for astronomy. This article presents original new optical observations from a variety of sources that indicate that one particular satellite (Blue Walker 3) - and others like it that are highly likely to follow - periodically becomes much brighter than IAU recommendations. The other major finding presented is the apparent inaccuracy of the TLE-derived positions of the satellites over time. The first result is quite clear, the second is less so in the presentation.

Originality & Significance: The work is original and significant, and should be published after recommended revisions - in particular, revisions to the included figures.

Data & Methodology: The presentation of the data and methodology in the supplement were very helpful. The results for the magnitude of BW3 as observed from around the globe are clearly presented, but need a more thorough presentation of the errors in the published magnitudes. For the deviations from the TLE positions of the satellites, it seems that the majority of the trend line (e.g. Fig. 4d) arises from a single dataset/instrument, and gives me less confidence that this trend line for large deviation from expected positions is supported by the data. I have no doubt that updated positions need to be reported regularly, but I am less confident that these data support that claim.

Appropriate use of statistics and treatment of uncertainties:

Please see comments in the line-keyed list below. Several of the recommended edits pertain to treatment of uncertainties in particular.

Conclusions: The conclusions about the variation in magnitude as a function of time during the deployment of BW3 are original and valid. The conclusions about the deviations from the TLE-derived positions as a function of time are less convincing.

Suggested improvements: Detailed descriptions of suggested improvements (from minor copy editing to recommended revisions in the text and figures) are listed below.

References: This is a very new area of study, and the authors have taken care to reference the appropriate existing works.

Clarity and context: The abstract, introduction and conclusions are clear and thorough, and this

paper is a meaningful contribution to the sparse literature available on this topic. In particular, it establishes a valuable example of establishing "ground truth" measurements of satellites and satellite constellations as they are being deployed.

Suggested Improvements:

Below is a list of questions, comments and suggested improvements keyed to line numbers in the PDF provided.

L 45 semicolon needed

L 49 Phased array antenna - what are the frequencies (see below)?

L 57 this - these processes

L 58 location predictions (on what timescale)?

L 63 "in the foreseeable future" - please give a timescale (yrs)

(Statistics - magnitude error bars should be discussed)

L 102 After the original posting (which is apparently delayed), how often are these databases updated with satellite telemetry?

L 107 The article should list the relevant downlink frequencies used by satellites, and if they are close to relevant protected RA bands (e.g. 10.68-10.7 GHz)

L 110 Add SKA, ngVLA, ALMA (e.g. radio observatories)

L 123 Starlink updates times are given. Are these frequent enough? Also, how often are BW3 telemetry data updated?

L130 mandatory impact assessments for satellite operators _prior to launch_

L 147 This application has been withdrawn (and replaced with another application). The correct - current - application should be referenced.

L 156 Figure 1 - magnitude error bars in the figure should also be referenced when magnitude values are given in the paper. "Fit" to data (green curve) may not be relevant since the changes in magnitude are not the result of any periodic process.

L 169 Figure 2. It would be helpful to label the trajectories with the dates listed in the figure caption.

L 177 Figure 3 A scale showing visual magnitudes on this plot (like Figure 2) would be helpful and relevant, as the values are also referenced in Figure 1.

L177 Figure 3 The FOV appears to be different in the top right figure (from CTIO)? Scale marking

should be added to the images, and the sub images should be labeled (a) to (f) for easier reference.

Comments on Methods section

L 218 tracking in sidereel -> tracking in sidereal

L262 PDF error? "values in the Θ - plane"

L271 Again, it would be nice to use this calculated value in a scale for Figure 3 "The magnitude of the trail was estimated"

L306 How were error bars for magnitudes calculated from Steward?

L322 "As the TLE did not match, we made a Gaussian fit to the normalised counts to spatially offset the TLE onto the trail centre." This could use a more thorough explanation.

L326 What statistical error is introduced in this photometric transformation?

L341 Is this the sole source for the calculated precision of TLE values?

L364 What is the source of these error bars?

L463 The decay rate is potentially significant - 1 arcminute per hour. Some more discussion of which of these 2 decay rates is more relevant would be helpful.

L470 Referring to "the middle panel" (second panel?) is confusing, since there are 4 panels. Labeling this subimages (a) to (d) would be helpful. Also confusing in this figure: the data/colors should be explained in the caption. (e.g. which data are the DDOTI data?) It is hard to interpret this plot without knowing which data correspond to which color. It would (perhaps) be helpful to use the same markers as used in Fig. 1, since these colors should be a subset of those? This would helpfully unify the figures.

L470 Presumably the arguments in this section result in the 7.2 ± 3.1 arcmin (mean total error from predicated position) in L116. The derivation of that value should be shown more explicitly in the Methods section.

Figure 4 The differing X-axis in (a) to (c) versus (d) is confusing and would be made more clear if these were separate figures.

Referee #2 (Remarks to the Author):

This paper includes observations of a new extremely large, bright satellite by astronomical facilities worldwide. The observations are aggregated over time, all scaled by the range to the satellite to show brightness at zenith, and show clear changes in the average brightness as the satellite

unfolded. The launch of this ultra-bright satellite by a small company has terrifying implications for the future of ground-based astronomy, and because the satellite launchers are not doing their own ground-based observing, it is left to the astronomy community to do this. This is extremely important, and although I strongly believe the satellite companies should be the ones funding this type of work, they are not doing it at present so all we can do is carefully document and disseminate what is happening to the sky. This work is absolutely important to users of the night sky worldwide.

The data presented are fairly straightforward astronomical observations, collected by a variety of talented teams on telescopes worldwide, and appear quite sound. I have a couple of suggestions below for improving presentation of some of the data, but overall, it looks great to me. The paper is well-written and I only have a few minor suggestions to improve clarity in a few places.

I have one fairly straightforward bit of analysis/visualization I would like to see implemented before publication (part 1), a few suggestions on information that I feel would be useful for Nature readers (who may not be quite as familiar with astronomy) to have included in this paper (part 2), and a few minor comments that would improve the clarity of the text (part 3).

Part 1: One more plot, please.

Figure 1 is clear and easy to read, but I am a little concerned that phase angle has been ignored here. I would really like to see another version of this plot (perhaps in the "Methods" section) that includes the solar phase angle information in some way. Figure 2 does somewhat show how magnitude changes with phase angle for a set of observations (and it's a beautiful figure!), and in the text you state that "range is the primary contributor to the apparent/observed magnitude," but you also state that "The apparent brightness of BW3 also shows correlation with solar phase angle." I would really like to see this in a plot: perhaps date on the x-axis, phase angle on the y-axis, magnitude shown by point size, and range shown by point colour? This particular example I've given might be too messy, but with the plots that are given in the manuscript currently, I don't have a good feel for if any observing sites actually measured a directly-on-zenith, high phase angle pass, or if the brightest points shown in Figure 1 are only bright because of scaling. Or perhaps there is enough information to reconstruct a sort of phase curve? That could also be a very interesting way to present this data. An additional plot that incorporates the phase angle information in some way would make the relative contributions of phase angle and range easier to understand at a glance.

Part 2: A few suggestions for additional background information to add.

It would be good to point out somewhere in the text that this major observing campaign took time and resources away from scientific research worldwide, which is generally funded by taxpayers, and the launching company did not compensate astronomers or taxpayers for that loss of science time.

It would also be good to emphasize the company's complete lack of communication with astronomers - it is my understanding that the vast majority of astronomers were completely surprised (~a few weeks out from launch) by the announcement from AST that this giant satellite

was coming soon. I'm sure you are aware that several of the major sat operators have at least sent representatives to attend the SATCON and DQS meetings, so it's quite worrying that a small company can launch a really astronomically-disruptive satellite with no prior communication.

Line 129 - It would be fair to point out here that while Starlink did attempt to mitigate the brightness of their satellites with a single "darksat" and with sunshades on many, they are no longer doing this and all of their gen2 satellites are going to be bigger than gen 1 (my point being: they have the ability to make their satellites fainter but are choosing not to, which is important contextual information for readers to know. Sigh.)

Part 3: Minor comments

-I suggest splitting sentence #2 in the abstract into 2 sentences, between "6" and "satellite operators"

-Suggest changing "some" to "many" in line 63 (the number I can observe with my naked eye in my sky definitely counts as "many")

-Line 85: "elevations" here is somewhat ambiguous - can you confirm/clarify that you mean "elevation of BW3 above horizon" rather than "elevation of observer above sea level"?

-line 100: this is perhaps off-topic, but is this also a problem for satellite-satellite collision avoidance if the LVA's independent orbit was not listed for 4 days?

Methods section:

Another additional plot suggestion: I would love to see a summary plot of all of the observing locations on the Earth as compared with the orbit of (or range of latitudes overflowed by) BW3.

Would that be possible to add? It would be helpful for knowing at a glance what BW3 phase angles are even possible to observe from different locations.

Referee #3 (Remarks to the Author):

Summary of the Key Results:

This paper presents new photometric and astrometric measurements of a novel, prototype, design of satellite intended to operate in Low-Earth Orbit. The paper finds that this new prototype, BlueWalker 3, is frequently so bright that it is clearly visible to the unaided eye, and therefore exceeds the IAU recommendations for such equipment.

Originality and significance: No observations of this new satellite design have been published previously, since this prototype is the first one launched. The paper outlines the challenges that large networks of such satellites could pose to optical and radio astronomy. The launch of this generation of extremely bright satellites to Low-Earth Orbit is a significant new development in the field.

Data and methodology, statistics:

Line 122: The decay rate quoted here is at the upper end of that actually measured, according to information in the Methods section of the paper, and is the result of including observations made at low elevation which deviate significantly from the rest of the data from a single site. Excluding those observations results in a much smaller rate, and the authors should acknowledge this in the main text.

Figure 1: I recommend that the authors adjust the color selections used for the different datasets to be more easily distinguished (e.g. cyan and green are too similar). I also recommend that a grid be used across the plot so that the reader can more easily read the values for particular observations. In the caption, the authors should better explain the placement of the middle vertical line, which they label as the “beginning of dimming”. The line does not coincide with any plotted data, so how was this transition determined?

Figure 2: I recommend that the grid squares be labeled with the intervals in degrees, and the green U-shaped cut-out next to the “West” label also have a label of its own to clarify the intended meaning (standardized solar azimuth?). Also, consider distinguishing the different tracks of points obtained on different days, either by using a different color-scale or with annotations.

While these plots are helpful, can the authors include a plot showing the brightness observations plotted as a function of elevation (standardized by solar azimuth as for Figure 2), with data from different nights distinguished by color and point-type? This would better illustrate the change in brightness of the satellites as they move relative to the Sun, and would make it easier to see the amplitude of second-order changes in the brightness, for example due to spacecraft attitude. This would also better justify the author's assertion on line 85.

Method: BW3 was trailed, even in the short exposure imaging, and in most of the images shown, there is no start or end point to the trail. This means that the satellite entered the field of view at some point during a finite exposure and left it, prior to the end of the exposure. This means that the flux measured across the trail is the sum of the flux over a shorter time period than the full

exposures. How did the authors compensate for this in their photometry? They compare the measured flux from the trail to the flux measured from background stars, but those stars were exposed for the full exposure time.

Minor comments

Throughout: The authors regularly refer to apparent visual magnitude for their measurements. While I infer from context that they are talking about measurements in the V passband, “visual magnitude” is unspecific, and this should be clarified, to clearly distinguish between measurements estimated by the human eye through binoculars, as presented in the section on “Visual observations”.

Specifically on line 74, the authors refer to observations made on 2022-11-10 which seem to be human-eye observations, and this should be made clear.

Line 65: The authors state that LEO satellite constellations pose challenges to cultural practices as well as to ground-based astronomy but do not provide a citation or arguments to justify this statement. I recommend they do both.

Line: 78-79: It might benefit the non-specialist reader if this paragraph included the approximate faintest magnitude star visible to the unaided eye from an inner-city and a dark site, so that they can compare this with the estimated brightness of the satellite.

Line 84: The authors should clarify the phrase “satellite-tracked observations” - this can be interpreted to mean an observation made by a satellite tracking some object, rather than (as I think the authors mean), observations made of the satellite by a telescope tracking at that satellite's rate of motion.

Line 91: How was the LVA later identified to be the second object, and by whom? If this was announced by the satellite operator for instance, then this should be stated.

Throughout the methods section, latitude and longitude measurements should indicate the units they are measured in.

Line 218: Correct spelling of “sidereal”.

Line 263: A symbol in the theta – something plane did not render properly in the PDF version of the manuscript that I received.

Line 347: “The observer has 20+ years of [experience] of tracking satellites...” Include missing word.

Figure 4: Are the symbols and color-coding used in this plot the same as for other figures? If so this should be stated in the caption; if not they should be explained. Again, adding a grid to the plotface would help the reader to evaluate individual measurements.

Author Rebuttals to Initial Comments:

Referee expertise:

Referee #1: radio interference

Referee #2: optical interference

Referee #3: optical interference

Referees' comments:

Referee #1 (Remarks to the Author):

Summary: The challenge of satellite constellations is an acknowledged issue for astronomy. This article presents original new optical observations from a variety of sources that indicate that one particular satellite (Blue Walker 3) - and others like it that are highly likely to follow - periodically becomes much brighter than IAU recommendations. The other major finding presented is the apparent inaccuracy of the TLE-derived positions of the satellites over time. The first result is quite clear, the second is less so in the presentation.

Originality & Significance: The work is original and significant, and should be published after recommended revisions - in particular, revisions to the included figures.

Data & Methodology: The presentation of the data and methodology in the supplement were very helpful. The results for the magnitude of BW3 as observed from around the globe are clearly presented, but need a more thorough presentation of the errors in the published magnitudes. For the deviations from the TLE positions of the satellites, it seems that the majority of the trend line (e.g. Fig. 4d) arises from a single dataset/instrument, and gives me less confidence that this trend line for large deviation from expected positions is supported by the data. I have no doubt that updated positions need to be reported regularly, but I am less confident that these data support that claim.

Appropriate use of statistics and treatment of uncertainties:

Please see comments in the line-keyed list below. Several of the recommended edits pertain to treatment of uncertainties in particular.

Conclusions: The conclusions about the variation in magnitude as a function of time during the deployment of BW3 are original and valid. The conclusions about the deviations from the TLE-derived positions as a function of time are less convincing.

Suggested improvements: Detailed descriptions of suggested improvements (from minor copy editing to recommended revisions in the text and figures) are listed below.

References: This is a very new area of study, and the authors have taken care to reference the appropriate existing works.

Clarity and context: The abstract, introduction and conclusions are clear and thorough, and this paper is a meaningful contribution to the sparse literature available on this topic. In particular, it establishes a valuable example of establishing "ground truth" measurements of satellites and satellite constellations as they are being deployed.

Suggested Improvements:

Below is a list of questions, comments and suggested improvements keyed to line numbers in the PDF provided.

L 45 semicolon needed

This sentence has been split in two per another referee's suggestion.

L 49 Phased array antenna - what are the frequencies (see below)?

These are now included.

L 57 this - these processes

We have removed this sentence as part of refactoring the summary paragraph.

L 58 location predictions (on what timescale)?

These were the TLE predicted sky positions, and a timing delay of 1.7s. We have removed this sentence as part of refactoring the summary paragraph.

L 63 "in the foreseeable future" - please give a timescale (yrs)

The sentence this refers to has been removed due to other feedback. It would otherwise have been changed to "during the 2030s." per Lawrence et al. 2022 (Ref. 1) - By 2030, potentially 100,000 objects (altitude of 600 km) are estimated.

(Statistics - magnitude error bars should be discussed)

We refer the referee to the Methods section for a discussion of how 1-sigma error bars are derived. We have expanded on this for observations that did not previously describe the uncertainties.

L 102 After the original posting (which is apparently delayed), how often are these databases updated with satellite telemetry?

This varies with object (and orbital type), but is roughly every 8 hours. We added a description of the LVA update frequency in the TLE Accuracy section.

L 107 The article should list the relevant downlink frequencies used by satellites, and if they are close to relevant protected RA bands (e.g. 10.68-10.7 GHz)

The downlink frequencies are 37.5-42.0 and 42.0-42.5 GHz (see FCC report <https://fcc.report/IBFS/SAT-PDR-20200413-00034/2266398>) which are above the protected RA bands at 10.68-10.7 GHz. Also these are used as downlink to user terminals: 890-894 MHz, 758-768 MHz. And for TT&C (telemetry, tracking and commands) 400.15-401 MHz. The protected RAS band close to these frequencies is not the 10.6-10.7 GHz band, but the

42.5-43.5 GHz. We have briefly mentioned these in the article, but we don't want to go into too much detail for an article that is focused on optical observations.

L 110 Add SKA, ngVLA, ALMA (e.g. radio observatories)

We added these.

L 123 Starlink updates times are given. Are these frequent enough? Also, how often are BW3 telemetry data updated?

We added a description of the BW3 update frequency in the TLE Accuracy section. As described in this section, the frequency of updates directly affects the positional uncertainties.

L130 mandatory impact assessments for satellite operators _prior to launch_

We added this.

L 147 This application has been withdrawn (and replaced with another application). The correct - current - application should be referenced.

Reference updated.

L 156 Figure 1 - magnitude error bars in the figure should also be referenced when magnitude values are given in the paper. "Fit" to data (green curve) may not be relevant since the changes in magnitude are not the result of any periodic process.

We have updated this figure, removing the green curve.

L 169 Figure 2. It would be helpful to label the trajectories with the dates listed in the figure caption.

We have added annotations with the date for each track.

L 177 Figure 3 A scale showing visual magnitudes on this plot (like Figure 2) would be helpful and relevant, as the values are also referenced in Figure 1.

We have included magnitude information for each streak in this Figure (now Figure 4).

L177 Figure 3 The FOV appears to be different in the top right figure (from CTIO)? Scale marking should be added to the images, and the sub images should be labeled (a) to (f) for easier reference.

The figure has been updated.

Comments on Methods section

L 218 tracking in sidereel -> tracking in sidereal

Changed.

L262 PDF error? "values in the Θ - plane"

Corrected to ' Θ -p plane'.

L271 Again, it would be nice to use this calculated value in a scale for Figure 3 “The magnitude of the trail was estimated”

As above, we estimated the magnitudes of each trail and added to the figure.

L306 How were error bars for magnitudes calculated from Steward?

The uncertainty was calculated with Astropy photutils tools (Ref. 25) following standard techniques. This is documented in the forthcoming Krantz et al. paper on “Steward Observatory LEO Satellite Photometric Survey”, and in the interest of conciseness, we do not explain it in detail here.

L322 “As the TLE did not match, we made a Gaussian fit to the normalised counts to spatially offset the TLE onto the trail centre.” This could use a more thorough explanation.

As is often the case, this TLE did not accurately predict a trail location. We therefore fit a Gaussian to locate the detected trail and measure the offset. We clarified this in the paper.

L326 What statistical error is introduced in this photometric transformation?

The introduced statistical error is negligible compared to the measurement uncertainty.

L341 Is this the sole source for the calculated precision of TLE values?

The overall TLE precision is based on both Leiden and DDOTI data. In addition, only about 22 arcseconds of the TLE precision is due to the astrometric accuracy of the camera. We now clarify this in the text.

L364 What is the source of these error bars?

We have expanded on this at the end of the visual observations section.

L463 The decay rate is potentially significant - 1 arcminute per hour. Some more discussion of which of these 2 decay rates is more relevant would be helpful.

We have removed the fit that included the low-elevation outliers, so we now just present one decay rate.

L470 Referring to “the middle panel” (second panel?) is confusing, since there are 4 panels. Labeling this subimages (a) to (d) would be helpful. Also confusing in this figure: the data/colors should be explained in the caption. (e.g. which data are the DDOTI data?) It is hard to interpret this plot without knowing which data correspond to which color. It would (perhaps) be helpful to use the same markers as used in Fig. 1, since these colors should be a subset of those? This would helpfully unify the figures.

We have reworked this figure so it now has labels a-d; changed the colours to correspond to Figure 1; and added legends.

L470 Presumably the arguments in this section result in the 7.2 ± 3.1 arcmin (mean total error from predicated position) in L116. The derivation of that value should be shown more explicitly in the Methods section.

We have clarified this point in the Methods section, the data is also shown more clearly in Extended Data Figure 2.

Figure 4 The differing X-axis in (a) to (c) versus (d) is confusing and would be made more clear if these were separate figures.

We tried having these as separate figures, but we did not think it worked as well as separate panels. We have improved these plots to make the scales clearer.

Referee #2 (Remarks to the Author):

This paper includes observations of a new extremely large, bright satellite by astronomical facilities worldwide. The observations are aggregated over time, all scaled by the range to the satellite to show brightness at zenith, and show clear changes in the average brightness as the satellite unfolded. The launch of this ultra-bright satellite by a small company has terrifying implications for the future of ground-based astronomy, and because the satellite launchers are not doing their own ground-based observing, it is left to the astronomy community to do this. This is extremely important, and although I strongly believe the satellite companies should be the ones funding this type of work, they are not doing it at present so all we can do is carefully document and disseminate what is happening to the sky. This work is absolutely important to users of the night sky worldwide.

The data presented are fairly straightforward astronomical observations, collected by a variety of talented teams on telescopes worldwide, and appear quite sound. I have a couple of suggestions below for improving presentation of some of the data, but overall, it looks great to me. The paper is well-written and I only have a few minor suggestions to improve clarity in a few places.

I have one fairly straightforward bit of analysis/visualization I would like to see implemented before publication (part 1), a few suggestions on information that I feel would be useful for Nature readers (who may not be quite as familiar with astronomy) to have included in this paper (part 2), and a few minor comments that would improve the clarity of the text (part 3).

Part 1: One more plot, please.

Figure 1 is clear and easy to read, but I am a little concerned that phase angle has been ignored here. I would really like to see another version of this plot (perhaps in the "Methods" section) that includes the solar phase angle information in some way. Figure 2 does somewhat show how magnitude changes with phase angle for a set of observations (and it's a beautiful figure!), and in the text you state that "range is the primary contributor to the apparent/observed magnitude," but you also state that "The apparent brightness of BW3 also shows correlation with solar phase angle." I would really like to see this in a plot: perhaps date on the x-axis,

phase angle on the y-axis, magnitude shown by point size, and range shown by point colour? This particular example I've given might be too messy, but with the plots that are given in the manuscript currently, I don't have a good feel for if any observing sites actually measured a directly-on-zenith, high phase angle pass, or if the brightest points shown in Figure 1 are only bright because of scaling. Or perhaps there is enough information to reconstruct a sort of phase curve? That could also be a very interesting way to present this data. An additional plot that incorporates the phase angle information in some way would make the relative contributions of phase angle and range easier to understand at a glance.

We have added a new Figure 2 that includes solar phase angle information for the initial data showing the high brightness of the satellite. In addition, to enhance clarity, the colour scheme in Figure 1 now also corresponds to that used in the Extended Data Figures.

Part 2: A few suggestions for additional background information to add.

It would be good to point out somewhere in the text that this major observing campaign took time and resources away from scientific research worldwide, which is generally funded by taxpayers, and the launching company did not compensate astronomers or taxpayers for that loss of science time.

We share the referee's concerns, and added an acknowledgement to this effect.

It would also be good to emphasize the company's complete lack of communication with astronomers - it is my understanding that the vast majority of astronomers were completely surprised (~a few weeks out from launch) by the announcement from AST that this giant satellite was coming soon. I'm sure you are aware that several of the major sat operators have at least sent representatives to attend the SATCON and DQS meetings, so it's quite worrying that a small company can launch a really astronomically-disruptive satellite with no prior communication.

Through the IAU CPS, many authors of the paper have communicated with industry, including AST. In fact, we chose to share an early draft of this manuscript with AST, and we incorporated their constructive comments prior to submitting it for publication. We share the referee's concerns and are actively engaging with many satellite operator companies.

Line 129 - It would be fair to point out here that while Starlink did attempt to mitigate the brightness of their satellites with a single "darksat" and with sunshades on many, they are no longer doing this and all of their gen2 satellites are going to be bigger than gen 1 (my point being: they have the ability to make their satellites fainter but are choosing not to, which is important contextual information for readers to know. Sigh.)

We are aware that Starlink is using Bragg coatings on their Gen2 satellites (we have added a mention of this into the sentence), and we at IAU CPS have an ongoing separate observational campaign to assess how well this has worked.

Part 3: Minor comments

-I suggest splitting sentence #2 in the abstract into 2 sentences, between "6" and "satellite operators"

The sentence has been split as suggested.

-Suggest changing "some" to "many" in line 63 (the number I can observe with my naked eye in my sky definitely counts as "many")

The sentence this refers to has been removed due to other feedback.

-Line 85: "elevations" here is somewhat ambiguous - can you confirm/clarify that you mean "elevation of BW3 above horizon" rather than "elevation of observer above sea level"?

clarified with 'of BW3 above the horizon'

-line 100: this is perhaps off-topic, but is this also a problem for satellite-satellite collision avoidance if the LVA's independent orbit was not listed for 4 days?

We don't know. It was only listed publicly after four days, but there are also private databases that are used for collision avoidance, so it may have been in that database much earlier.

Methods section:

Another additional plot suggestion: I would love to see a summary plot of all of the observing locations on the Earth as compared with the orbit of (or range of latitudes overflowed by) BW3. Would that be possible to add? It would be helpful for knowing at a glance what BW3 phase angles are even possible to observe from different locations.

We have added the requested plot as Extended Data Figure 1.

Referee #3 (Remarks to the Author):

Summary of the Key Results:

This paper presents new photometric and astrometric measurements of a novel, prototype, design of satellite intended to operate in Low-Earth Orbit. The paper finds that this new prototype, BlueWalker 3, is frequently so bright that it is clearly visible to the unaided eye, and therefore exceeds the IAU recommendations for such equipment.

Originality and significance: No observations of this new satellite design have been published previously, since this prototype is the first one launched. The paper outlines the challenges that large networks of such satellites could pose to optical and radio astronomy. The launch of this generation of extremely bright satellites to Low-Earth Orbit is a significant new development in the field.

Data and methodology, statistics:

Line 122: The decay rate quoted here is at the upper end of that actually measured, according to information in the Methods section of the paper, and is the result of including observations

made at low elevation which deviate significantly from the rest of the data from a single site. Excluding those observations results in a much smaller rate, and the authors should acknowledge this in the main text.

We have removed the second decay rate that was including the low elevation data, so we now just show the fit with the low elevation data excluded.

Figure 1: I recommend that the authors adjust the color selections used for the different datasets to be more easily distinguished (e.g. cyan and green are too similar). I also recommend that a grid be used across the plot so that the reader can more easily read the values for particular observations. In the caption, the authors should better explain the placement of the middle vertical line, which they label as the “beginning of dimming”. The line does not coincide with any plotted data, so how was this transition determined?

We have updated this figure, removing the green curve, adding a grid, and revising the colours of the markers. We have also made the vertical line descriptions clearer. About “beginning of dimming”: The date December 8, 2022 is taken from the reference [8] where the dimming of BW3 was discussed, and this reference is included in the image caption. It is provided as a dotted vertical line to show that the satellite has started to appear bright again.

Figure 2: I recommend that the grid squares be labeled with the intervals in degrees, and the green U-shaped cut-out next to the “West” label also have a label of its own to clarify the intended meaning (standardized solar azimuth?). Also, consider distinguishing the different tracks of points obtained on different days, either by using a different color-scale or with annotations.

We have revised this figure accordingly.

While these plots are helpful, can the authors include a plot showing the brightness observations plotted as a function of elevation (standardized by solar azimuth as for Figure 2), with data from different nights distinguished by color and point-type? This would better illustrate the change in brightness of the satellites as they move relative to the Sun, and would make it easier to see the amplitude of second-order changes in the brightness, for example due to spacecraft attitude. This would also better justify the author's assertion on line 85.

We have added a new Figure 2 that includes solar phase angle information for the initial data showing the high brightness of the satellite.

Method: BW3 was trailed, even in the short exposure imaging, and in most of the images shown, there is no start or end point to the trail. This means that the satellite entered the field of view at some point during a finite exposure and left it, prior to the end of the exposure. This means that the flux measured across the trail is the sum of the flux over a shorter time period than the full exposures. How did the authors compensate for this in their photometry? They compare the measured flux from the trail to the flux measured from background stars, but those stars were exposed for the full exposure time.

The referee is correct that the satellite trail flux is a function of exposure time. We have corrected all our reported trail magnitudes according to the exposure time and distance travelled by the satellite across the sky to determine the satellite's angular speed given its known altitude.

For images where there is no start of end points, a correction is required for the background star integrated flux, to take into account the shorter exposure time of the satellite. The flux of the background stars is scaled using the time ratio of transverse time divided by exposure time. The transverse time is found by taking the trail length and dividing by the angular velocity. This is an automatic process done by the CLEOSat pipeline. We have added a description of these corrections near the start of the Methods section, to better explain to the reader.

Minor comments

Throughout: The authors regularly refer to apparent visual magnitude for their measurements. While I infer from context that they are talking about measurements in the V passband, “visual magnitude” is unspecific, and this should be clarified, to clearly distinguish between measurements estimated by the human eye through binoculars, as presented in the section on “Visual observations”.

Specifically on line 74, the authors refer to observations made on 2022-11-10 which seem to be human-eye observations, and this should be made clear.

We have distinguished between measurements in the V passband and human eye observations by calling these “V-band” and “visual” throughout. In addition, we double checked that the uncertainties in the transformation between visual and V-band are below the measurement uncertainties.

Line 65: The authors state that LEO satellite constellations pose challenges to cultural practices as well as to ground-based astronomy but do not provide a citation or arguments to justify this statement. I recommend they do both.

The paragraph containing these points has been removed due to editorial feedback pertaining to introductory material.

Line: 78-79: It might benefit the non-specialist reader if this paragraph included the approximate faintest magnitude star visible to the unaided eye from an inner-city and a dark site, so that they can compare this with the estimated brightness of the satellite.

We have added a sentence giving these numbers.

Line 84: The authors should clarify the phrase “satellite-tracked observations” - this can be interpreted to mean an observation made by a satellite tracking some object, rather than (as I think the authors mean), observations made of the satellite by a telescope tracking at that satellite's rate of motion.

We have simplified this to 'Optical observations'. Most of the observers did not actively track the satellites.

Line 91: How was the LVA later identified to be the second object, and by whom? If this was announced by the satellite operator for instance, then this should be stated.

It was done retroactively through TLE matching after it became listed in the public catalog four days after the unfolding event. We have clarified this in the paper.

Throughout the methods section, latitude and longitude measurements should indicate the units they are measured in.

We have added the units (degrees).

Line 218: Correct spelling of “sidereal”.

Corrected.

Line 263: A symbol in the theta – something plane did not render properly in the PDF version of the manuscript that I received.

Corrected to ' Θ -p plane'.

Line 347: “The observer has 20+ years of [experience] of tracking satellites...” Include missing word.

Missing word added.

Figure 4: Are the symbols and color-coding used in this plot the same as for other figures? If so this should be stated in the caption; if not they should be explained. Again, adding a grid to the plotface would help the reader to evaluate individual measurements.

The figure has been updated, and now the colour-coding matches Figure 1. We also added a grid and legends to this figure.

Reviewer Reports on the First Revision:

Referees' comments:

Referee #1 (Remarks to the Author):

I made extensive comments on the first submission of this paper, so I will not reiterate those comments here.

I have carefully reviewed the authors responses, and I am convinced that they have done a thorough job of addressing the comments of myself and the other referees. I think that in particular the improvements made to the figures, and the addition of references to radio frequency impacts are very helpful.

One minor comment: there is a reference to a blue dotted line in Extended Data Fig. 2b (thank you for the subfigure references), and this reference should refer to Fig. 2b specifically. That is, caption should read:

"A linear trend is observed showing an accuracy decay rate of 0.4 ± 0.2 arcmin hr⁻¹, when the Takahashi data is excluded (blue dashed line, Extended Data Fig. 2b).

Referee #2 (Remarks to the Author):

I am very pleased with the changes that have been made incorporating referee comments - especially the new figures. I particularly like the color-coding that references between Figure 1 and Extended Data Figure 1. Great!

I have no additional changes and I recommend this for publication in its current form.

Thank you for doing this important work and I sincerely hope this paper gets a lot of interest and publicity from journalists!

Referee #3 (Remarks to the Author):

Summary of the key results

This paper presents new photometric and astrometric measurements of a new design of satellite intended to operate in Low-Earth Orbit. The paper finds that this new prototype, BlueWalker 3, is frequently so bright that it is clearly visible to the unaided eye, and therefore exceeds the IAU recommendations for such equipment.

Originality and significance:

As this is a prototype of a new class of satellite, the data and analysis presented in the paper is original, and provides a valuable real-world observations that will assist astronomers to evaluate the

impact of these satellites on astrophysics.

Data & methodology:

The author's have evaluated the optical brightness of the satellite by performing photometric measurements from the trails it makes through images acquired along the predicted orbital track. This 'wait and catch' approach has been successfully applied for such fast moving objects previously. The authors analyze data from some of the facilities using their new CLEOSat software for astrometric calibration and trail analysis, while other datasets were reduced with software appropriate to the data. Visual observations from an experienced observer are included with estimated uncertainties that scale appropriately with the availability of comparison stars.

Appropriate use of statistics and treatment of uncertainties:

The authors evaluate the measured brightness of the satellite as a function of time since launch and solar phase angle. Since the primary goal of the work is to assess the range of optical brightnesses reached by the satellite, their analysis and statistics are straight forward and appropriate.

Conclusions

Based on their observations, the authors conclude that this satellite exceeds the IAU guidelines, and highlight the potential detrimental impacts on astronomy of launching constellations of satellite this bright.

Suggested improvements:

I have no further suggestions, and I recommend this paper for publication.

References: appropriate

Clarity and context: The paper is well written and as concise as possible.

Author Rebuttals to First Revision:

We thank the referees for checking our manuscript again.

We have made the following changes based on the two comments.

One minor comment: there is a reference to a blue dotted line in Extended Data Fig. 2b (thank you for the subfigure references), and this reference should refer to Fig. 2b specifically. That is, caption should read:

"A linear trend is observed showing an accuracy decay rate of $0.4 \hat{\pm} 0.2$ arcmin hr⁻¹, when the Takahashi data is excluded (blue dashed line, Extended Data Fig. 2b).

We have made the change.

Referee 1 has a minor comment to be addressed. I have discussed the paper with the chief physical science editor, and the editor in chief of Nature. They both recommend that you omit the sentence at line 96-98, to avoid any possible legal ramifications. Also, there is the possibility that the application will be turned down, or the company may change plans. If either of those happen, then the paper could be out dated.

Based on this, we have removed the sentence that was on line 96-98.